Cells determine cell density using a small protein bound to a unique tissue-specific phospholipid

Petzold Christopher J. 1
Schwarz Richard I. 2 rischwarz@lbl.gov
1 Joint BioEnergy Institute, Lawrence Berkeley National Laboratory , Berkeley, CA , USA
2 Life Sciences Division, Lawrence Berkeley National Laboratory , Berkeley, CA , USA
Tobin Desmond
Electronic publication date: 2013 Oct 29
Publication date: 2013
Volume: 1
Electronic Location ID: e192
Received 2013 Jun 25; Accepted 2013 Oct 8
Copyright: © 2013 Petzold et al.
Copyright year: 2013
Copyright holder: Petzold et al.
License: This is an open access article distributed under the terms of the Creative Commons Attribution License, which permits unrestricted use, distribution, and reproduction in any medium, provided the original author and source are credited.
License URL: https://creativecommons.org/licenses/by/3.0/

Keywords: Cell density signaling, Tendon morphogenesis, Bone morphogenesis, Membrane signaling, Growth plate formation

Funding: CRADA BG92-106 Breast Cancer Research Foundation N.I.H. R37CA064786 This work was supported in part by a CRADA BG92-106, N.I.H. grant R37CA064786, and a grant from The Breast Cancer Research Foundation. The funders had no role in study design, data collection and analysis, decision to publish, or preparation of the manuscript.

==============================
Cell density is the critical parameter controlling tendon morphogenesis. Knowing its neighbors allows a cell to regulate correctly its proliferation and collagen production. A missing link to understanding this process is a molecular description of the sensing mechanism. Previously, this mechanism was shown in cell culture to rely on a diffusible factor (SNZR [sensor]) with an affinity for the cell layer. This led to purifying conditioned medium over 4 columns and analyzing the final column fractions for band intensity on SDS gels versus biological activity – a 16 kD band strongly correlated between assays. N-terminal sequencing – EPLAVVDL – identified a large gene (424 AA), extremely conserved between chicken and human. In this paper we probe whether this is the correct gene. Can the predicted large protein be cleaved to a smaller protein? EPLAVVDL occurs towards the C-terminus and cleavage would create a small 94 AA protein. This protein would run at ∼10 kD, so what modifications or cofactor binding accounts for its running at 16 kD on SDS gels? This protein has no prominent hydrophobic regions, so can it be secreted? To validate its role, the chicken cDNA for this gene was tagged with myc and his and transfected into a human osteosarcoma cell line (U2OS). U2OS cells expressed the gene but not passively: differentiating into structures resembling spongy bone and expressing alkaline phosphatase, an early bone marker. Intracellularly, two bands were observed by Western blotting: the full length protein and a smaller form (26 kD). Outside the cell, a small band (28 kD) was detected, although it was 40% larger than expected, as well as multiple larger bands. These larger forms could be converted to the predicted smaller protein (94 AA + tags) by changing salt concentrations and ultrafiltering – releasing a cofactor to the filtrate while leaving a protein factor in the retentate. Using specific degradative enzymes and mass spectrometry, the bone cofactor was identified as a lipid containing a ceramide phosphate, a single chained glycerol lipid and a linker. Tendon uses a different cofactor made up of two fatty acid chains linked directly to the phosphate yielding a molecule about half the size. Moreover, adding the tendon factor/cofactor to osteosarcoma cells causes them to stop growing, which is opposite to its role with tendon cells. Thus, the cofactor is cell type specific both in composition and in the triggered response. Further support of its proposed role came from frozen sections from 5 week old mice where an antibody to the factor stained strongly at the growing ends of the tendon as predicted. In conclusion, the molecule needed for cell density signaling is a small protein bound to a unique, tissue-specific phospholipid yielding a membrane associated but diffusible molecule. Signal transduction is postulated to occur by an increased ordering of the plasma membrane as the concentration of this protein/lipid increases with cell density.

Introduction

To function correctly, a cell needs a real-time assessment of its environment. A key component of this running tally is cell density – a parameter known to alter both the rates of cell proliferation and the level of cell differentiation. Loss of cell density awareness has long been associated with the onset of cancer and the ability of malignant cells to overgrow the monolayer when grown in cell culture (Aaronson & Todaro, 1968). Similarly, in normal cells the slowdown in growth as cells become dense is associated with the induction of differentiated function (Kaldis & Richardson, 2012). Despite its importance to normal function, how a cell “sees” its neighbors remains an enigma. The sensing mechanism is known to be influenced by high concentrations of growth factors, for instance high serum levels, but this has not shed light on how a cell actually detects the presence of its neighbors (Pardee, 1987; Vogel, Ross & Raines, 1980). Therefore, to better comprehend tissue morphogenesis and stability requires an understanding of how cells signal their presence to each other.

Tendon morphogenesis is highly dependent on cell density signaling. This is because the tissue is over 90% type I collagen and the two parameters controlling local collagen deposition – the rate of cellular collagen production and the cell number – are cell density regulated (Rowe & Schwarz, 1983; Schwarz & Bissell, 1977). Cell density regulation of cell proliferation and apoptosis causes a growth plate to form, allowing the cells to deposit an even distribution of collagen along the longitudinal axis of the fibril – a hallmark of this tissue (Schwarz, 1996).

In cell culture primary avian tendon (PAT) cells from 16 day embryos devote ∼50% of their total protein production to procollagen (Schwarz, Farson & Bissell, 1979). This high level is identical to that seen in ovo where tendon development occurs rapidly (∼11 days) enabling the newly hatched chick the ability to walk. To produce high levels of procollagen from a single copy gene and allow rapid regulation puts restrictions on where this pathway can be controlled. Transcription is an unlikely candidate because induction is slow from a single copy gene requiring 3 days to fully induce procollagen mRNA levels in PAT cells (Rowe & Schwarz, 1983). Moreover, the procollagen mRNA is stable (24 h half-life) so returning to the uninduced state can take over 2 days (Lyons & Schwarz, 1984). Instead, PAT cells regulate procollagen at a post-translational step. Translation and secretion rates are both tied to formation of a triple helical molecule (Rowe & Schwarz, 1983; Schwarz, 1985) and this in turn requires hydroxylation of prolines to stabilize this conformation. The enzyme, prolyl hydroxlase, responsible for this rapid regulatory control has two subunits and the level of the alpha subunit is dependent on cell density (Kao, Kao & Schwarz, 1985; Lee, Kao & Schwarz, 2001).

To begin to understand cell density regulation, PAT cells were grown as a 6 mm island in the middle of a 60 mm dish (Schwarz, 1991). The cells in middle of the island grow to be confluent while cells at the edge of the island would grow outwards to be at low cell density. In this way, cells at multiple cell densities could be studied at the same time. Indeed, when one probed the levels of procollagen mRNA by in situ hybridization, the level dramatically increased from the edge of the island to the confluent center (Schwarz, 1991). Growing cells as an island was useful but it made a change that turned out to be much more significant: the ratio of medium to cells was increased 100-fold. While gently agitating PAT cells confluent over the whole dish had no affect, gently agitating PAT cells grown as an island caused dramatic changes by increasing cell proliferation and decreasing procollagen production (Schwarz, 1991; Zayas & Schwarz, 1992). The reason for this difference is that cells confluent over the whole dish could condition the medium such that shaking only increased the rate at which the factor exchanged locations between the medium and the cell layer and not its concentration. When grown as an island of cells, there was too much medium and the factor released from the cell layer was basically lost. The lower concentration of this factor strongly altered the ability of PAT cells to sense the cells around them.

Adding shaken conditioned medium (CM) from confluent cultures would strongly stimulate an island of PAT cells to grow and this became an assay to purify this factor. Using 400 liters of CM as starting material, this was serially applied to 4 columns. Analyzing the 30 fractions from the last column (reverse phase), one set was run on SDS gels and the second set was analyzed for biological activity. The intensity of the band running at 16 kD tracked best with activity. This band was sequenced and yielded: EPLAVVDLTEKTIS (Schwarz, 2002). This chicken sequence does not have an immediate correspondence to the human equivalent for two reasons. First, as with all N-terminal Edman sequence data, with each cycle the noise level increases reducing the confidence of each additional read. Not unexpectedly, the chicken genome shows an exact homology only with the first 8 AA. Second, the human genome has a variant in the fifth position from V to E. Nevertheless, this 8 letter AA sequence in the chicken comes from a much larger gene with a gene product of 424 AA from seven exons. The homology between the chicken and the human genes at the amino acid level is 89% identical and 94% positive. For this large gene to be correct requires further processing of the protein product to a smaller form: a specific cleavage (there is no methionine for an alternative start site) at the EPLAVVDL site would give the right starting sequence; however, this sequence is found near the carboxy end and would only produce a protein with 94 AA, ∼10 kD. So to run at 16 kD on SDS gels would require that the protein be modified or that the protein bind a cofactor. In addition the 94 AA cleavage product does not contain highly hydrophobic regions that could facilitate the secretion to the outside of the cell.

In this paper, a candidate gene for the diffusible cell density signal is tagged with myc and his tags and ectopically expressed in another cell type from another species. We probe whether the expressed gene gets cleaved, whether the cleaved product gets secreted, and the modification or cofactor that allows a 94 AA protein run at 16 kD. Finally, we want to test whether the CM containing the recombinant factor when applied to an island of PAT cells acts like the original diffusible cell density signal. This classic test is complicated by the finding that the cell type producing the 94 AA protein binds a unique tissue-specific lipid cofactor and this active cell density signal imparts a tissue-specific response.

Materials and Methods

Cells and cell culture

PAT cells were isolated from 16 day chick embryos by a modification of the procedures described by Dehm & Prockop (1971). Specifically, a higher serum level (3%) was used in the dissociation medium and the use of a nylon mesh (20 µm; TETKO) instead of filter paper to separate out single cells. In these experiments the medium was changed from F12 to a 50/50 mix of F12/DMEM in order to be consistent with the medium used to grow the U2OS cells. The medium contained 0.2% fetal bovine serum deactivated for 30 m at 56°C.

U2OS cells (from frozen stocks that have been in the laboratory for over a dozen years (Campeau et al., 2009)) were grown in F12/DMEM medium with 0.2% fetal bovine serum. In 75 cm2 flasks the medium, 50–75 ml, was changed every other day. When the cells first overgrow the monolayer, they are very sensitive to medium volume and will apoptose if too much medium is added.

CM was produced by growing cells in 225 cm2 flasks. The medium was changed to serum-free medium and the volume reduced to 12 ml. The flasks were placed in an environmental rotator at 100 rpm, 37°C for 1 h. The CM was collected and new medium added, and the process was repeated for the third hour. After the 3rd collection medium with serum was added (50–75 ml) and the flask returned to the regular incubator.

The 16 day chick embryos were exempt from requiring an animal use protocol as determined by LBL Animal Welfare and Research Committee. Similarly, the legs from 5 week mice (BALB/c) came from mice already euthanized for other approved protocols. This is also an exempt use.

Alkaline phophatase assay

The Sigma kit was used (85L2-1KT) and the included procedures were followed with only slight modifications for labeling tissue culture flasks instead of slides.

Genes, cloning, and expression

RNA was extracted from confluent cultures of PAT cells using Trizol (Invitrogen). The RNA was reversed transcribed using superscript II (Invitrogen). The cDNA was amplified by PCR using proof reading taq polymerase (Invitrogen) using primers from the 5′ end of exon 2 plus a methionine start site and the 3′ end of exon 7.

Forward primer: GCTGGATCCATGAAAGACCGTCTCAACCTTCCA

Reverse primer: ACAGAATTCGACTTCATCCTGGGCTCC

This was cloned into the vector pcDNA3.1/Myc-His(+)A,B,C (Invitrogen) and sequenced. The sequence agreed with the predicted expression of this gene (minus exon 1; accession: XP_427094). Using primers for standard T7 promoter/priming site and the BGH Reverse priming site with the addition of a Sal1 site, the gene with tags was amplified using PCR. It was TOPO cloned (Invtrogen) to enhance Sal1 activity and then cut with BamH1 and Sal1. This was ligated to pESY-Neo-II (Koh, Chen & Daley, 2002; Lee, 2007), a retroviral vector, that had also been cut with BamH1 and Sal1.

In summary, at the time the sequence for the chicken exon 1 was not known, so a 3 AA sequence containing methonine was added to the 5′ end (above the dashed line below) and myc and his tags added to the 3′ end (below the dashed line below). This was inserted into a retroviral vector pESY-Neo-II – retroviral pEYK 3.1 was modified to remove the GFP and bleomycin DNA and replace it with Neo/Kan antibiotic resistance gene and a multi-cloning site (Lee, 2007) and renamed pESY1c:

M K D ————————————————————————————–————————————————————————————–————–————–————–————–————–————– R L N L P S V L V L N S C G I T C A G D E N E I A A F C A H V S E L D L S D N K L E D W H E V S K I V S N V P H L E F L N L S S N P L S L S V L E R R C A G S F A G V R K L V L N N S K A S W E T V H T I L Q E L P D L E E L F L C L N D Y E T V S C S P V C C Q S L K L L H I T D N N L Q D W T E I R K L G I M F P S L D T L I L A N N N L T T I E E S E D S L A R L F P N L R S I N L H K S G L H C W E D I D K L N S F P K L E E V K L L G I P L L Q S Y T T E E R R K L L I A R L P S I I K L N G S I V G D G E R E D S E R F F I R Y Y M E F P E E E V P F R Y H E L I T K Y G K L E P L A V V D L R P Q S S V K V E V H F Q D K V E E M S I R L D Q T V A E L K K H L K T V V Q L S T S N M L L F Y L D Q E A P F G P E E M K Y S S R A L H S Y G I R D G D K I Y V E P R M K ————————————————————————————–————————————————————————————–————–————–————–————–————–————– S N S A D I Q H S G G R S S L E G P R F E Q K L I S E E D L N M H T G H H H H H H

Alternating normal and bold/italic fonts indicate exons and the underlined AAs in last exon indicate the secreted form of the gene (94 AA).

The vector pESY1c (4 µg/60 mm2 dish) was used to transfect phoenix cells using lipofectamine 2000 using the manufacturer’s protocol (Invitrogen). The cells were incubated overnight and the next day medium was collected, filtered (0.45 µm), and used to transfect pESY1c into U2OS cells in the presence of 8 µg/ml polybrene. The transfected cells were selected using G418 (100 µg/ml).

Polyclonal antibody

The secreted form of pEsy1c (underlined AA above, 94 AA) with a methionine at the N-terminus (above dashed line) was inserted into the E. coli expression vector pTrcHis2-TOPO (Invitrogen). The vector included myc and his tags at the C-terminus (below lower dashed line).

M ————————————————————————————–————————————————————————————–————–————–————–————–————–————– E P L A V V D L R P Q S S V K V E V H F Q D K V E E M S I R L D Q T V A E L K K H L K T V V Q L S T S N M L L F Y L D Q E A P F G P E E M K Y S S R A L H S Y G I R D G D K I Y V E P R M K ————————————————————————————–————————————————————————————–————–————–————–————–————–————– D D D D K K G E F E A Y V E Q K L I S E E D L N S A V D H H H H H H

Forward primer: GTGACCATGGAGCCCTTGGCAGTCGTGGAT

Reverse primer: CTTATCGTCATCGTCCTTCATCCTGGGCTC

The procaryotic expression vector yielded an expressed protein that was 7 AA smaller in the linker between the protein and the tags (the variation is a function of the type of multiple cloning site and the primers used in cloning) and 1 AA larger at the N-terminus. This difference was too small to detect on gels and the linker region was not important for antibody production. The protein was expressed in E. coli and purified and nickel columns (Invitrogen). This protein (4 mg) was used to immunize rabbits and for affinity purification (Biosynthesis). This protein was also used as a standard for the secreted form of the factor with tags.

Immunofluorescence

Cell culture: cells were fixed with 4% paraformaldehyde for 10 m in a high calcium buffer (.05 M TES pH 7.2, 0.1 M NaCl, .05 M CaCl2). The fixation was quenched with two, five minute washes in buffer A (.05 M TES pH 7.2, 0.1 M NaCl, .01 M CaCl2) containing 0.1 M glycine. A blocking agent (1% casein Hammerstein [EM Science] in PBS) was used for 30 m followed by a rinse with buffer A. The primary antibody was a polyclonal made in rabbits against the secreted form of pESY1c plus tags expressed in E. coli. This was added at 1:40 (the antibody was diluted by 25% with stabilizer [Candor]) for 2 h with agitation. Then, washed 3X with buffer A with agitation and then a fluorescently tagged goat anti-rabbit secondary was added for 1 h with agitation. Washed 3X with buffer A and then once with a DAPI buffer (500 nM in PBS). Finally, a wash with buffer A.

As a control, a preimmune serum from the same rabbits was used at 1:40 (the serum was diluted 25% with stabilizer) and used as described above.

Tissue sections: hind legs from 5 week old mice were dissected to reveal the muscle and the attached tendon. These were embedded in OCT (Sakura) for 10 to 30 m and then frozen in small embedding molds in a dry ice/ethanol bath. Frozen tissue was sliced into 16 µm sections and attached to superfrost plus slides (VWR). The sections on the slide were outlined with a pap pen and then analyzed as described above. In this case, because the tendon had less autofluoresence in the red, this was the preferred color for the secondary emission. Because the plane of the knife and the plane of the tendon were infrequently in agreement, the analysis required many legs and sections.

Western blot analysis

For cytoplasmic proteins: cells were lysed (400 µl/75 cm2) with a detergent buffer (150 mM NaCl, 1% NP-40, and 50 mM Tris [pH 8.0]). This was diluted 2X sample buffer (Invitrogen) and treated as described below.

For extracellular protein: samples were concentrated on 10 kD spin ultrafilters (Millipore) diluted with 2X sample buffer and concentrated again to <200 µl. The samples were made 5% β-mercaptoethanol and heated at 80°C for 10 m. 40 µl or less was applied and run on an SDS gel (4–20%, 1.5 mm, 10 well, Tris-Glycine [Invitrogen]). The gels were blotted onto nitrocellulose (0.45 µm), washed with water, and dried overnight. Next, the blots were rehydrated with water and then blocked with 1% Hammerstein casein in PBS for 1/2 h. After 2X water washes the primary antibody (monoclonal myc (Millipore) or polyclonal rabbit, see above) was added at 1:1000 in PBS with 0.2% casein Hammerstein for 2 h. The blot was then washed 3X for 5 m in PBS with .02% Tween 20. After which the rabbit secondary antibody conjugated to horseradish peroxidase (Thermo) was added at 1:1000 in PBS with 0.2% casein. Again the blot was washed 3X for 5 m in PBS with .02% Tween 20 and then developed with a Dura or Femto kits (Thermo). The blots were photographed with a FluoroChem HD2 (ProteinSimple).

Antibody selection and specificity

Antibody affinity to a protein could be strongly affected by a lipid cofactor. This became an issue when the monoclonal antibody (clone 4A6, Millipore) to the myc tag stopped working. Over the course of these experiments Millipore changed the storage buffer several times and finally to a liquid stored at 4°C. This latter change reduced the affinity of the antibody so that only the free protein factor was detected (data not shown). The benefit of the antibody from the 4A6 clone is that it “recognizes and is specific for recombinant proteins containing the Myc epitope tag in a variety of sequence contexts”. Other clones are known to show strong affinity differences depending on the protein attached to the myc tag. Add a lipid cofactor to a protein with a myc tag and it is unclear how a context sensitive antibody would bind. To find an adequate replacement, three alternatives were tested: a rabbit polyclonal antibody to the secreted form of the factor (94 AA + tags) expressed in E. coli; a monoclonal antibody to the myc tag from Applied Biological Materials (clone A7) and one from GenScript (clone 2G8D5). Using the same CM sample for each antibody yields a Western blot that is unique to the probing antibody (Fig. 1). Critical to our purification procedure for the cofactor, only the polyclonal antibody can detect the free factor at our test concentration of ∼0.5 ng as well as bands with the factor/cofactor. Most importantly, one cannot compare the intensity of one band to that of another because each antibody has its preferred conformations; instead, one can only compare the relative amount of a specific band after different treatments.

Figure 1 The conformation of the tagged factor and its interaction with the cofactor dramatically influences the detection by monoclonal and polyclonal antibodies.

During the course of these experiments Millipore changed the stabilizing agent for its monoclonal antibody (clone 4A6) and it would now only recognize the free form of the factor. The Millipore monoclonial antibody had been selected because it detects the myc sequence in a “variety of sequence contexts.” Two other monoclonal antibodies were tested – Genescript clone 2G8D5 (lanes 1–5) and ABM clone A7 (lanes 6–10) – to see if they could be an adequate replacement. In addition, a polyclonal antibody made in rabbits to the secreted form of the factor (94 AA + tags) expressed in E. Coli was tested (lanes 11–16). For each case, the same samples were used: MW markers (lanes 1, 6, 11), the retentate from trying to separate the factor from the cofactor at 1X (lanes 2, 7, 12) and 2X (lanes 3, 8, 13, 14), and the free factor with tags made in E. Coli at 2X (∼1 ng, lanes 4, 9, 15) and 1X (∼0.5 ng, lanes 5, 10, 16). The monoclonal antibodies show no overlap in what they detect and neither of them detects the free factor made in E. Coli at the concentration used. The ABM monoclonal is striking in how strong it reacts to only one form at ∼75 kD with almost no background. The polyclonal does prefer the ∼50 kD form, shows some affinity for the 75 kD form as well as weak affinity for some of the lower MW forms. The polyclonal antibody is sensitive to the secreted factor made in E. coli and can be used in an assay for the separation of the factor from the cofactor. The critical point is that one cannot compare the intensity between bands but only the change in intensity of one band after different treatments.

Cofactor purification

The CM was filtered through a 0.2 µm filter and then the NaCl level was increased by adding 1/10 the volume with 5 M NaCl. The high salt CM, 10 to 50 ml, was concentrated to 1 ml in a stirred cell (10 ml; Millipore), at 700 rpm and pressurized (14 psi, N2). The filter was a YM-10 with a nominal cutoff at 10 kD (Millipore). The retentate was diluted to 10 ml with water and reconcentrated to 0.5 ml. The retentate was analyzed on Western blots to see the level of free factor released.

The filtrate containing the cofactor was concentrated using a Pierce C18 spin column (Thermo). The column was wetted with 1 ml of 50% acetonitrile/water and then spun for 1m at moderate speed in a clinical centrifuge. Then, 1 ml of the filtrate was added and spun and repeated until all the filtrate was bound to the column. After which the column was washed twice with 1 ml of water, and twice with 1 ml of acetonitrile. Then, the washing was repeated: 1 ml of 50% acetonitrile, 1 ml water, 2X 750 µl acetonitrile, and eluted with 2X 750 µl of chloroform. Samples were stored in chloroform and concentrated before analysis by blowing down with nitrogen to ∼100 µl. For mass spectrometry, 10 µl of sample was diluted with 80 µl of acetonitrile and 10 µl water. All solvents were lc, ms, or hplc grade.

With recently purchased YM-10 filters (in comparison to using older filters from the original maker Amicon), the cofactor would not separate from the factor unless the following changes were made: the filters had to soak at 37°C in 20% acetonitrile for 48 h; after concentrating the high salt CM to 1 ml, it was put into 10 ml of buffer (10 mM EDTA[pH 8.0], 2 ml acetonitrile, 3 ml isopropyl alcohol, 5 ml water) and heated to 80°C for 2 h and then concentrated to 0.5 ml using the treated YM-10 filter. The filtrate was further concentrated and purified as described above.

Lipid enzyme treatment

Lipid degradative enzymes were first used to characterize the makeup of the cofactor. In this case, we were looking for a change in the mobility of factor/cofactor on a Western blot after being treated with these enzymes for various times 30 m, 90 m, 180 m, and a no enzyme control. CM (3 ml) was concentrated to 0.5 ml (one sample for each time point). Then the enzymes were added as detailed below and incubated for the specific time, 2X sample buffer was added to stop the reaction, concentrated on a 10 kD spin ultrafilter, and analyzed by Western blotting.

Lipase: Sigma (L8525); 380 units/µl; added 50 µl; incubated 37°C in buffer B (see below)

Phospholipase C: Sigma (P7633); .038 units/µl; added 50 µl; incubated 37°C in buffer B

Phospholipase A2: Sigma (P9279); 6 units/µl; added 50 µl; 20 mM triethanolamine pH 8.9 substituted for TES in buffer B; incubated room temperature.

Sphingomyelinase: Sigma (S9396); 0.54 units/µl; added 2 µl; incubated 37°C in buffer B with MgCl2 substituted for CaCl2.

The second use of lipid enzymes was to identify the peaks on mass spectrometry that were sensitive to the enzyme treatment. In the cofactor purification above, between the wash steps, the column material was resuspended in buffer B (20 mM TES pH 7.2; 10 mM NaCl; 1 mM CaCl2), enzyme added, and incubated for 30 min at 37°C with agitation. This was spun down and the protocol above continued. In the buffer, when using sphingomyelinase, CaCl2 was replaced with MgCl2.

Mass spectrometry analysis

The diluted lipid extract samples were analyzed on an ABSciex TripleTOF 5600 (AB Sciex, Foster City, CA). Samples were introduced to the mass spectrometer via syringe pump by using a Nanospray III source (ABSciex) with a nano-tip emitter (New Objective, Woburn, MA) operating in positive-ion mode (2400 V). The data were acquired with Analyst TF 1.5.1 by averaging the signal from 400 m/z to 1250 m/z over several minutes. MS/MS spectra were collected in “high sensitivity” mode with the quadrupole set to UNIT resolution and collision energy set to 30 to optimize fragmentation. MS/MS spectra were scanned from 100 m/z to 1600 m/z and were collected for a total accumulation time of 500 ms.

Results

Expression in U2OS cells causes differentiation

A replication defective retroviral vector containing a neomycin resistance gene was used to transfect the chicken gene plus tags into U2OS cells (Koh, Chen & Daley, 2002; Lee, 2007). Because this study was started when the chicken genome was first sequenced, the chicken equivalent to the first exon was not present. So the insert contained only the last six exons (pESY1c).

U2OS cells, derived from a human osteosarcoma tumor, showed stable expression of this gene after infection when grown in low serum media (0.2%) (Valmassoi & Schwarz, 1988). Despite the low serum levels, U2OS cells transfected with pESY1c would start to overgrow the monolayer as small ridges. The ridges would coalesce into circles sharing boundaries with other circles that resembled spongy bone in a two dimensional space (Fig. 2B). Uninfected U2OS cells showed some ability to overgrow the monolayer but could not maintain the growth needed to form completed structures (Fig. 2A). The multicell ridges produced alkaline phosphophatase on the cell surface, indicating increased expression of the osteoblast phenotype (Fig. 2C).

Figure 2 Expression of the cDNA, pESY1c, causes human osteosarcoma cells (U2OS) to differentiate.

Control U2OS cells will overgrow the monolayer but cannot sustain growth necessary to form structures (A). U2OS-pESY1c cells, after becoming confluent (end of week 1), the cells begin to overgrow the monolayer. First as linear short stripes but these coalesce into small circles and then finally some of the common borders recede and the circles become larger (week 3, phase contrast, B). When U2OS-pESY1c cells get to the stage of differentiation where they are making large circles of overgrown cells, they start to produce alkaline phosphatase, an early marker of bone development. In this assay using brightfield microscopy, the overgrowing cells turn dark, an indicator of alkaline phosphatase activity, while the confluent cells inside the circle express little or no activity (C). Scale bars = 100 µm.

Pattern of molecular expression intracellularly and extracellularly

A monoclonal myc antibody and Western blotting was used to probe the expression and processing of this myc tagged gene. Two specific bands were seen intracellularly: ∼44 kD band which is expected for the full size gene product; ∼26 kD band that could be a cleavage product (Fig. 3). However, this band is larger that the predicted 20 kD protein (16 kD + tags).

Figure 3 Western blot analysis of U2OS-pESY1c expression both inside and outside the cell.

The monoclonal antibody to the myc tag (clone 4A6) detects c-myc inside the cell so control U2OS cells were used to identify the background bands (lanes 1, 2). For each condition, two lanes were run on an SDS gel where the second was loaded with twice the sample. U2OS cells expressing pESY1c showed two additional bands – a full length protein at 44 kD and a smaller band at 26 kD (lanes 3, 4). To analyze what was secreted from the cell, the CM was put through a 30 kD ultrafilter and the retentate and the filtrate were further concentrated on a 10 kD ultrafilter so that a sufficient level of protein could be loaded onto the gel. The retentate shows a wide spectrum of bands from the smallest at 28 kD, the most intense at ∼60 kD and the highest at ∼250 kD (lanes 5, 6). The filtrate showed one band at ∼60 kD (lanes 7, 8). Since the filtrate initially went through a 30 kD ultrafilter, this large band is an indication that concentrating the sample causes aggregation.

To observe the secreted forms a simple purification was used whereby the CM was put through a 30 kD ultrafilter. Both the retentate and the filtrate were analyzed and both had to be further concentrated on a 10 kD ultrafilter in order to load sufficient protein on the gel for Western blot analysis (Fig. 3). The retentate contained many bands: the smallest was a weak band at ∼28 kD, the strongest was at ∼60 kD, and the largest was over 200 kD. The filtrate showed basically a single band at ∼60 kD. This was unusual since 60 kD proteins are excluded to a high degree from passing through a 30 kD ultrafilter. One explanation is that by concentrating the sample ∼20-fold causes the tagged protein to aggregate into stable forms that resist the denaturing effects of heat, β-mercaptoethanol, and SDS.

Aggregation is due to a cofactor

The multiple bands seen in the Western blot complicates the interpretation of how this tagged protein is working in the extracellular space. But more specifically, even the smallest MW band (∼28 kD) seen outside the cell is higher than predicted. Is this larger size due to a different amount or type of modification in bone cells versus tendon cells? Or is this increase in apparent MW a reflection of an interaction with another molecule?

To begin to answer these questions, we needed to resolve how the unmodified secreted form of the protein factor with two tags (94 AA + tags) migrates on a Western blot. To accomplish this, the tagged protein was expressed in E. coli, purified on a nickel column, and then used as a standard in Western blots. The tagged secreted form of the protein ran with a MW of 16 kD (Fig. 4, lane 6). This confirms that there is either a covalent modification or a binding to another molecule that gives rise to the 28 kD form in U2OS cells. Further aggregation could then occur when concentrating the factor to give rise to larger MW forms.

Figure 4 Determining whether the apparent aggregation of the factor in the CM is due to covalent modifications or binding of a cofactor.

When the CM was concentrated using a 10 kD ultrafilter and this sample was analyzed by Western blot using a myc antibody (clone 4A6), one sees a lane of multiple bands, the smallest being 28 KD (lane 1). In contrast, if the secreted form of the factor with tags (94 AA + tags) is expressed in E. coli, one sees a single band at 16 kD (lane 6). Lane 5 was reserved for MW markers. The CM was put through a 30 kD ultrafilter and the retentate used as a control (lane 4) and the filtrate was divided into two parts. The first was further concentrated on 10 kD ultrafilter and then made 30% DMSO in order to see if this solvent could affect the mobility of the factor (lane 2). The DMSO did limit the aggregation in that more of the factor ran at 28 kD. In contrast, when the filtrate was treated with high salt (0.5 M) and then low salt by 10 fold dilution and then concentrated on a 10 kD ultrafilter, the bands almost all went to a 16 kD form and its 32 kD dimmer (lane 3). This is strong evidence that whatever was causing the increased MW to 28 kD and the further aggregation was not covalently bound.

This raises the question of whether different physical or chemical environments could alter the size of the bands observed on Western blots. CM from U2OS-pESY1c was put through a 30 kD ultrafilter and the filtrate was then concentrated on a 10 kD ultrafilter. This was divided into 2 parts. One was further concentrated on a 10 kD ultrafilter for loading on a gel but in this case, 30% DMSO was added. To the other sample, salt was added to raise the concentration by 0.5 M; concentrated with an ultrafilter; then diluted to low salt; and then concentrated to apply to a gel (Fig. 4). The effect of DMSO was to reduce the number of bands, especially those at high MW, but the smallest band still remained 28 kD. In contrast, the treatment with high salt, low salt and ultrafiltration yielded a much simpler picture where the bands were reduced to the 16 kD band and its dimer at 32 kD. One can conclude that the 28 kD band is due to a binding of a cofactor and not a covalent modification. In addition, when the cofactor is separated from the secreted form of the protein factor, the MW of the protein is the expected 16 kD (94 AA + tags). This shows that U2OS cells are capable of cleaving the large gene product to the 94 AA + tags and then secrete it.

Localization and characterization of the factor/cofactor in the extracellular space

The previous experiment raises the question of how the untagged 16 kD form in tendon cells compares to the 28 kD tagged form seen in bone cells. Even taking into account a 4 kD tag, there is an additional 8 kD shift in MW. Are these forms variations with common characteristics or is the bone form acting in a different manner? In tendon cells the 16 kD form is loosely bound to the cell layer. This gives a key feature to this form in that gently shaking a flask of tendon cells causes them to loose their ability to sense high cell density, resulting in increased proliferation.

First, the localization was examined with immunofluorescence using a rabbit polyclonal antibody specific to the tagged secreted form of the chicken protein expressed in E. coli. During tests to optimize the signal, procedures that disrupted membranes decreased the signal. Using an optimized protocol, immunofluorescence staining was localized to areas where the cells have overgrown the monolayer (hills) and expression was suppressed in monolayer areas (valleys) (Fig. 5A).

Figure 5 Localization of the bone factor/cofactor in the extracellular space and disruption by gentle agitation.

U2OS-pESY1c cells were grown in culture for 9 days and then fixed with 4% paraformaldehyde in a high calcium buffer. A rabbit polyclonal antibody was used as a primary and goat anti-rabbit alexa 488 was used as a secondary (A). For comparison, the same field is shown in phase contrast (B). Areas where the cells are overgrowing the monolayer show strong staining whereas cells remaining in the monolayer appear to be suppressed. This leads to the strong localized differences in the expression of this gene. When CM was collected from these cells, the flasks are gently rotated (100 rpm) for 1 h; then the medium is collected and new medium is added. After 3 collections the flasks are stationary for the next 21 h. After collecting medium for 3 days a dramatic change is observed in the culture – cells that had overgrown the monolayer begin to spread into regions that had remained a monolayer (C). Scale bar = 100 µm.

Second, the cells were subjected to gentle shaking (100 rpm) for 3 h per day with the medium changed each hour. After 3 days the hills had spread into the valleys and the valleys had additional growth. The net effect was to even out the growth over the flask (Fig. 5C).

One can conclude that some of the major physical properties of the bone and tendon factor/cofactor are the same. They are both localized to the cell layer – most likely bound to the cell membrane. However, this interaction is not strong, allowing significant diffusion that can be enhanced by gentle agitation of the medium.

Characterization of the bone cofactor with degradative enzymes

The cofactor is tightly bound to the factor and gives a consistent mobility pattern on Western blots. This pattern could then be used to test whether specific degradative enzymes could cleave the cofactor, alter the migration profile, and reveal the partial composition of the cofactor.

One possible candidate for the cofactor would be a variation on the anchoring structure for membrane bound proteins: a glycosylphosphatidylinositol lipid (Paulick & Bertozzi, 2008). To test for a glycolipid, the factor/cofactor was treated with a variety of polysaccharide degradative enzymes – chondroitinase ABC, hyaluronidase, heparinase – and the pattern on a Western blot was compared to that of an untreated control. None of the treated samples showed a distinct change in the pattern (data not shown). At the same time, a variety of lipid degrading enzymes were tested – lipase, phospholipase A2, phospholipase C, and sphingomyelinase. Lipase, phospholipase C, and sphingomyelinase caused dramatic shifts in the pattern seen on Western blots (Fig. 6). Lipase was the most dramatic causing bands to shift to lower MW including the 16 kD, free protein form. The lipase was a crude preparation used at a high concentration; so while its primary target would be a glycerol fatty acid linkage in the cofactor, this is probably not the only cleavage product. The phospholipase C used in this experiment has overlapping activities with the sphingomyelinase. The sphingomyelinase is more specific to cleaving moities on ceramide phosphates. In contrast, phospholipase A2 did not show a significant change in mobility or intensity (data not shown). A negative result is hard to interpret because there are too many potential causes. The positive results, while not yielding a definitive structure, implicate lipid components containing a phosphoceramide and a glycerol fatty acid as part of the cofactor. The lipid nature of the cofactor can explain why the factor/cofactor has a strong tendency to aggregate when concentrated due to hydrophobic interactions.

Figure 6 Treating U2OS-pESY1c CM with various lipid degradative enzymes to identify the makeup of the bone cofactor.

The only assay for the cofactor is its binding to the factor and shifting its mobility. The CM was treated with lipase, phospholipase C, and sphingomyelinase for 30, 60, and 180 min and analyzed by Western blots using a myc monoclonal antibody (A, lanes 1, 2, 3; A, lanes 7, 8, 9; and B, lanes 2, 3, 4; respectively). The E. coli expressed secreted form of the factor (94 AA + tags) was run as a control (A, lane 5 showing a monomer band at 16 kD and a dimer band at 32 kD) as well as untreated CM (A, lane 6; and B, lane 1). A, lane 4 contained MW markers (data not shown). The lipase was the least specific enzyme and it degrades the cofactor and frees most of the factor to run at 16 kD. This type of phospholipase C can cleave at both glycerol phosphates and ceramide phosphates. The sphigomyelinase is specific for ceramide phosphates. Both of these enzymes cause single breaks and this alters the binding/mobility but does not free the cofactor from the factor.

Purification and further characterization of the bone cofactor using mass spectrometry

In order to characterize the bone cofactor in more detail, procedures were needed to separate both the factor from the cofactor and then the cofactor from other lipid-like molecules. The first step in the purification – the separation of the cofactor from the factor – could benefit from standard lipid procedures using a methanol/chloroform extraction. In this procedure the lipids are found in the chloroform layer and proteins are found as a precipitate between the layers. There is no assay for the cofactor but the proteins in the precipitate can be solubilized and then Western blot analysis can be used to see if the protein factor runs as the free form (16 kD) or as a complex with the cofactor (28 kD). The amount of free factor is an indicator of cofactor release. However, redissolving the protein precipitates at the interface and running Western blots only showed the 28 kD form and higher order aggregates (Fig. 7). In this extraction protocol, the cofactor remained bound to the factor, and as a consequence, precipitated at the interface.

Figure 7 Chloroform/methanol extraction does not free the lipid bone cofactor from the protein factor.

To test whether the standard extraction method of purifying lipids would yield a free cofactor, CM from U2OS-pEsy1c was extracted with chloroform/methanol. The protein fraction which precipitates at the interface between the two solvents was examined by Western blotting using a myc monoclonal antibody (lane 1 is half the sample volume of lane 2). Despite unusual sample preparation, the SDS gels and Western blotting worked reasonably well. Nevertheless, there was no evidence of a 16 kD free protein factor. Instead the bands resembled those of the factor/cofactor running together with a minimum size of 28 kD.

Despite attempts to weaken the interaction between the factor and the cofactor by changes in pH, salt, solvent, chaotropic agents, temperature, and chelating agents, Western blots showed little or no conversion to the free form (data not shown). The only successful method required a combination of a shift in salt concentration and ultrafiltration (Fig. 4). Using a simplified version of this procedure whereby the CM was made high in salt (∼0.6 M) and then concentrated on a 10 kD ultrafilter. This was then diluted 10-fold with water and concentrated to run on Western blots (Fig. 8). By intensity of the bands, most of the tagged factor runs as the free form. Thus, the corresponding free cofactor was expected to have passed through the ultrafilter as part of the filtrate.

Figure 8 A simplified procedure using ultrafiltration and high salt releases the bone cofactor.

In this composite Western blot (using a myc monoclonal antibody, clone 4A6) where the molecular weight markers (lane 2) are superimposed on the other lanes. The secreted form of the factor made in E. coli (94 AA + tags) was run in lane 1 and the concentrated CM was run in lanes 3, 4. To release the cofactor the CM was made ∼0.6 M NaCl and then concentrated on a 10 kD YM-10 ultrafilter and then diluted with water and refiltered using the same filter (lanes 5, 6). In this case, the retentate shows a strong shift to the secreted form of the factor without cofactor, 16 kD.

Using a small C18 resin column to concentrate, differential solvent elution to purify, and then comparing the untreated and the enzyme treated samples via mass spectrometry one can identify candidate cofactor peaks (Fig. 9). Only molecular weights between 530 and 580 are shown because there was no significant differences between the samples at either higher or lower values (mass range = m/z 300 – m/z 1500). Two doubly-charged ions at m/z 543.4273 and m/z 557.4429 were dramatically reduced by treatment with either enzyme and these peaks were not observed in the buffer only control and the intensity of the m/z 557 ion was highly correlated with the intensity of the free protein factor in the retentate. The observed difference between the doubly-charged ions was 14 representing a 28 Da difference in MW commonly seen for lipids. In the rest of the analysis, we focused on the 557 species with a MW of 1114.8858. By using a high mass accuracy Q-TOF mass spectrometer we generated potential molecular formulas with a list of atoms commonly found in organic biomolecules (i.e., carbon, hydrogen, nitrogen, oxygen, and phosphorous). The most likely formula C63H123N2O11P (exact mass = 557.4432252+; 0.6 ppm from the measured mass) had nearly a 1:2 Carbon:Hydrogen ratio common to lipids and the nitrogen, oxygen and phosphorous indicative of a hydrophilic region. Support for the proposed cofactor structure derived from MS/MS fragmentation of m/z 557. Two fragment ions, shown in Fig. 10, at m/z of 323 and m/z 378 were consistent with a ceramide phosphate and a linker moiety, supporting the results from the enzymatic reactions.

Figure 9 Identification of the bone cofactor on mass spectrometry by using lipid degradative enzymes and differential elution from a C-18 spin column.

After separation from the factor, the bone cofactor was in the dilute filtrate. To concentrate the lipid containing cofactor, the filtrate was run over a C-18 column and washed with water, acetonitrile, and eluted with chloroform. To identify the cofactor, the cofactor was treated separately with phospholipase C (B) and lipase (C). These were compared to untreated cofactor (A) and a lipase buffer only control (no cofactor; D) using mass spectrometry. The only regions that showed a strong peak in the untreated sample and little or no intensity in the other panels were at an m/z of 543.42 and 557.44. These peaks had the characteristic pattern of a doubly charged molecule and this gave the cofactor a MW of 1086.84 or 1114.88.

From our analysis, the cofactor was susceptible to three lipid enzymes – lipase, phospholipase C, and sphingomylenase, and a peak observed at m/z 557 that has a strong lipid composition. This peak was lost when treated with lipase, phopholipase C, and sphingomylenase. Using our knowledge of which enzymes cleave the cofactor and the molecular formula, a working model can be postulated that includes a ceramide phosphate, a glycerol lipid, and a linker consisting of a short modification of a proline residue between them (Fig. 10).

Figure 10 Working model is shown for the bone cofactor.

Using our knowledge of the lipid enzyme preferences, the molecular weight of the whole molecule, the predicted molecular formula, and the fragmentation pattern (MS/MS), a preliminary model has been drawn with a ceramide phosphate, a single chain glycerol lipid, and a small linker. This model also explains why this molecule is soluble from aqueous buffers to chloroform: a structure with a hydrophilic end and hydrophobic tails. The model also shows how the fragmentation pattern of two breaks (MS/MS) is consistent with the model.

Tendon cofactor characterization using enzymes and mass spectrometry

Having analyzed the bone factor/cofactor, the focus returned to the original aim of characterizing the tendon factor, but now knowing that it binds a cofactor. The bone and tendon cofactors must be different because they impart a unique shift in mobility to the factor/cofactor complex on SDS gels. This raises important questions about the nature of the tendon cofactor and how it compares to its bone counterpart.

As a first step in the characterization, the same panel of lipid degradative enzymes were tested on PAT cell CM. In this case, concentrated CM was used that was over 6 months old. This results in a stable aggregate at ∼60 kD being the dominant band. As with bone, lipase and phospholipase C were active on the PAT cell factor/cofactor while phospholipase A2 was inactive (Fig. 11A). Unlike the bone cofactor, sphingomyelinase was inactive on the PAT cell factor/cofactor (Fig. 11B). To insure that the inactivity of the enzyme was not due to the factor/cofactor being in a specific aggregate, the treatment with sphingomylinase was repeated with fresh CM (Fig. 11C) and the result remained unchanged. This indicates a lack of a ceramide phosphate moiety in the tendon cofactor.

Figure 11 Characterizing the tendon cofactor by its sensitivity to specific lipid degradative enzymes.

PAT cell CM was put through a 30 kD filter and then concentrated with a 10 kD filter. This concentrated CM after several months at 4°C forms a dominant aggregate that runs at ∼60 kD. This CM was run on Western blots (using a polyclonal antibody) as follows: untreated (A, lanes 2, 6); lipase treated (A, lanes 3, 7); phospholipase A2 (A, lanes 4, 8); and phospholipase C (A, lanes 5, 9). The data indicates a strong sensitivity to lipase and phospholipase C but not to phospholipase A2. This is an identical sensitivity pattern as seen with the bone cofactor. Next sphingomylinase was tested (B, lanes 5, 6, 7) and compared to the untreated control (B, lanes 2, 3, 4). In contrast to the bone cofactor, sphingomyelinase did not affect the tendon cofactor. To rule out that the sphingomyelinase was being inhibited by the aggregation caused by “aging” the CM, fresh cultures of PAT cells were made and the CM treated with sphingomyelinase (C, lanes 4, 5) and untreated control (C, lanes 2, 3). Again, this enzyme does not alter the mobility of the tendon factor/cofactor.

The cofactor was purified for mass spectrometry by binding it to C18 columns followed by washing with methanol, acetonitrile, and elution with chloroform. Via mass spectrometry we compared preparations that were treated with either phosphospholipase C or lipase while the cofactor was bound to the C18 column to the untreated cofactor samples. No differences were seen in the methanol or acetonitrile washes; however, the chloroform fraction did reveal several peaks that changed between the untreated and treated samples. Two singly charged peaks, at 668.61 and 696.65, were reduced to background levels by the enzyme treatments. Similar to the bone cofactor mass spectrometric analysis, these peaks differed by 28 MW and this reflects a variation in lipid chain length by two carbons. Additionally, there was no doubly charged ion observed at m/z 557 as was seen in the bone cofactor sample (data not shown).

Using MS/MS to further characterize the 668.61 ion yielded a rather simple spectrum with two peaks of roughly equal intensity and with MW of 283.26 and 311.29. These ions differ by 28 and accurate mass measurements correspond to saturated fatty acids with 18 and 20 carbons, respectively. Both chains are required to be in the model in order to achieve the high lipid content indicated by the parent molecule’s MW. Since the band is degraded by phospholipase C, there is a phosphate group yet the total mass of the intact ion is insufficient to include a commonly observed glycerol phosphate. Consequently, our proposed structure includes anhydride linkage of the fatty acid directly to the phosphate. The mass difference between the total mass and the fatty acid and phosphate groups corresponds to a cyano group, thus our working model adds a cyano group to the phosphate and is consistent with the enzymology, the mass spectrometry data, and the gel mobility data (Fig. 12). Whether the cyano group was part of the original molecule or it displaced another group during the purification remains to be determined.

Figure 12 A working model is shown for the tendon cofactor.

The same approach as previously used with the bone cofactor yielded two peaks at 668.61 and 696.65 separated by 28. This difference is commonly seen for lipid chains by addition of two CH2 groups. Since the 668 band was stronger, we focused the model on this band. The ms/ms fragmentation pattern was fairly simple yielding two bands of roughly equal intensity corresponding to 18- and 20-carbon fatty acids. Since the molecule is degraded by phospholipase C, a phosphate was added to the model. However, there was insufficient MW for a glycerol so the fatty acids were attached directly to the phosphate. The remaining MW fits a cyano group. The working model is presented to give a framework to the type of structure expected.

Nomenclature

With two different cofactors and the potential for more in the future, there is a need to give them distinct names. For the bone factor/cofactor will be named SNZR (sensor)1P1L; for tendon factor/cofactor will be named SNZR 1P2L. Similarly, the protein factor will be SNZR 1P and the bone cofactor, SNZR 1L and the tendon cofactor, SNZR 2L. At present, because of the extremely high homology between species at the protein level and no knowledge of the conservation of lipid structure between species, the delineation is only on cell type.

Cofactor tissue specificity

The cofactor for bone and the one for tendon are unique. The SNZR 1P can bind either SNZR 1L or SNZR 2L and drive a unique differentiation pathway dependent on the type of cofactor produced by the cell. This raises the question of whether the SNZR 1P2L when added to U2OS cells is capable of driving bone differentiation or does it elicit a new or null response? In addition, one can ask whether the protein factor, by itself, has any activity?

To answer these questions, U2OS cells were grown in regular media as a control, or with 5X concentrated condition media from U2OS-pESY1c cells expressing the SNZR 1P1L, and from 5X concentrated condition media from PAT cells expressing SNZR 1P2L. In addition, SNZR 1P purified from E. coli, was added at 5 µg/ml – 3 orders of magnitude greater than the estimated level in CM (Figs. 13A–13D). These levels were chosen because early work with PAT cells have shown that 1X CM is insufficient to stimulate growth when added back to confluent cultures (Zayas & Schwarz, 1992). The high level of the protein factor was chosen because it seemed unlikely that the factor without the cofactor would have a lot of activity and small amounts of activity are difficult to detect by visualization.

Figure 13 The cofactor has a unique lipid makeup depending on the producing cell type but would SNZR 1P1L and SNZR 1P2L give the same or different response if added to type?

In this case we used the control cell line U2OS and added a 5X concentrated condition medium from PAT cells or from U2OS-pESY1c cells. An additional test was done to see if the free factor made in E. coli had any activity on U2OS cells if applied at a high concentration (5 mg/ml). In comparison to expressing pESY1c in U2OS cells which causes hills and valleys to form, adding 5X CM from these cells to uninfected U2OS cells shows only a slight growth effect after 6 days as the cells appeared to be more compact (B) vs medium only control (A). Growing the cells for an additional 9 days showed a strong growth effect (F) vs the medium only control (E). Moreover, the increased growth is even over the culture – no hills or valleys. In contrast, adding 5X CM from PAT cell cultures caused the U2OS cells to grow more slowly and increased the presence of more rounded up cells (D). Adding high concentrations of the secreted form of the factor alone (expressed in E. coli, 94 AA + tags) did not appear to increase growth after 6 days (C). Scale bar = 100 µm.

After a week in the various media, the most dramatic effect was that the tendon CM was inhibitory to growth when added to the U2OS cells yielding a subconfluent culture. The addition of the SNZR 1P alone did not appear to change the cultures compared to the control even at this high concentration. Adding 5X CM from U2OS cells expressing the SNZR 1P1L, showed some increased in growth as the cells appeared to be more compact. However, to confirm this observation we allowed this flask and its control to grow for 9 additional days, at which point it was obvious that the CM had caused the cells to completely overgrow the monolayer (Figs. 13E, 13F). What is particularly striking is the uniformity of growth when the CM is added to cells versus the lattice patterns that are formed when the cells themselves express the SNZR 1P1L (Fig. 2B).

Tissue localization in mice

One aim of this paper – producing the recombinant form of the tendon cell density signal in another cell type and then applying it to tendon cells to show it has the original activity – is not compatible with our data. Since the cofactor that is essential for activity is unique (in two cases) to the cell type producing it, another test would be useful in showing that the protein factor is the correct candidate for being part of the cell density signaling mechanism. There is the similarity that both the tendon SNZR 1P2L and bone SNZR 1P1L are loosely bound to the cell layer and cause the cells to differentiate. In addition, the polyclonal antibody to the 94 AA protein does detect a band at 16 kD in the PAT cell CM (Fig. 11C). A more functional test would further confirm the identity.

To accomplish this, the polyclonal antibody was used to check whether the factor was at the tendon/muscle junction of 5 week old mice. This is the growing stage for mice and one would predict that the factor would be at high concentration at the growing ends of the tendon where there should also be a high density of fibroblasts (Schwarz, 1996). To detect the protein factor, a polyclonal rabbit antibody was used with a secondary antibody labeled with alexa 568; to detect the nuclei, the sections were stained with DAPI (Fig. 14).

At the growing end of the tendon there is a strong fluorescence signal for the factor (Fig. 14A) as well as numerous nuclei labeled with DAPI (Fig. 14B). There is variation in the signal over the length of the tendon but this can result from the plane of the tendon not being in exactly the same plane as the section. The strong staining for the factor at the growth plate of the tendon supports its role as a cell density signal.

Figure 14 Localization of the factor using a rabbit polyclonal antibody in 5 week old mice at the tendon/muscle junction.

In the first photograph (A) a small tendon is seen from the lower left edge of the image towards the middle. The polyclonal antibody staining for the factor (using a secondary antibody, goat anti rabbit alexa 568), showed intense staining compared to the muscle bundles. Staining the same section with DAPI shows that this part of the tendon is full of nuclei reflecting that this is a growing end (B). The connective tissue surrounding the muscle shows nuclei staining with DAPI that looks like a line of nuclei. The light staining of the muscle connective tissue with the polyclonal antibody suggests another cell density signal with its own cofactor. As a control, a preimmune serum was used at the same concentration and exposed for the same time, was used to detect non specific signal (C). The same section was DAPI stained and the stained nuclei show a growing tendon rising steeply from the lower left corner to the top of the picture (D). One has to take into account that the tendon does not have to be in exactly the same plane as the tissue slice and therefore, the nuclei density can vary as the tendon weaves through the muscle. Scale bar = 100 µm.

As a control, a preimmune serum was used at an equal concentration and exposed for equal time and it is basically negative (Fig. 14C) and the DAPI staining shows the tendon nuclei starting at the lower left corner and rising steeply to the top of the picture (Fig. 14D).

One unexpected observation is the light staining of the muscle connective tissue with its single file nuclei (Figs. 14A and 14B). This is an indication that a third cell type may be producing this protein factor along with its own unique cofactor.

Discussion

In 1976, we showed that freshly isolated chicken embryonic tendon cells were exquisitely sensitive to cell density as reflected in the levels of procollagen production (Schwarz, Colarusso & Doty, 1976). Despite this dramatic regulation of the most important protein in tendon morphogenesis, the paper went on to say that the “mechanism by which one cell communicates its presence to another cell is unclear”. A few years and many experiments later the signaling mechanism is now shown to use a tight complex between a protein and a lipid. The protein is a small cleavage product from the carboxy end of a larger gene product and this small protein is subsequently secreted from the cell. The protein itself appears to have little or no activity but requires the binding of a lipid cofactor. Remarkably, the composition of the lipid is tissue-specific. While only two types have been characterized, they are each unique in both the lipid structure and the triggered response.

SNZR 1P is also unique in composition, processing, and function. By being able to bind unique phospholipids, SNZR 1P allows diffusion into the medium of a lipid that would otherwise be expected to be tightly embedded in the membrane. In earlier papers, this trait was described as being “loosely bound to the cell layer” an explanation for its ability to localize to the cell layer but be partially removable by gentle agitation (Schwarz, 1991; Stoker, 1973; Zayas & Schwarz, 1992). Being able to identify SNZR 1P and its lipid partners gives this complex a molecular identity that explains its unusual characteristics. In addition, as more has become known about the SNZR 1P1L, similarities can be seen to other signaling molecules – in particular, sonic hedgehog. There is no sequence homology but they are both made as large proteins that are processed to a smaller form and then secreted from the cell (Weed, Mundlos & Olsen, 1997). In the case of sonic hedgehog, it is the N-terminus that is secreted and in this case, it is the C-terminus. The activity of sonic hedgehog is greatly enhanced by the covalent linkage of palmitate and/or cholesterol (Taylor et al., 2001); with the cell density signaling the protein itself has little activity but this is greatly enhanced by binding to a lipid cofactor. Because of the lipid attachments, sonic hedgehog runs as aggregates on SDS gels (Goetz et al., 2006). The binding of the cofactor causes a similar effect on the SNZR 1P. On the other hand, the differences are equally important: SNZR 1P can bind a variety of lipids and in doing so can change its signaling function in a tissue specific manner. Besides how this complicates the analysis of gene function, the result is that the cell has the ability to sense both the type and number of cells in its immediate vicinity and thereby react accordingly. Given the nature of the complex, the simplest hypothesis is that the protein holds the lipid in a specific conformation that can then bind to the cell membrane and organize it (Simons & Toomre, 2000). With more cells there would be additional binding resulting in a more organized plasma membrane and this would trigger a stronger signal transduction.

But we should point out that our focus has been on the carboxy end of the transcript yet the rest of the protein is equally well conserved between the chick and the human. Another group searching for homology to other tubulin stabilizing proteins found a 20% overlap with this gene (Bartolini et al., 2005). By Northern analysis, many tissues appeared to transcribe this gene at low levels while testis produced it in larger amounts. Over expressing this gene in HeLa cells caused tubulin to be destabilized in contrast to their initial selection criteria. Because they tagged the transcript at the amino terminus, the secreted form would have gone undetected. Moreover, they pointed out that they could not distinguish whether the destabilization of tubulin was a direct or an indirect effect. If the effect on tubulin was an indirect effect of cell density signaling in HeLa cells, then the conservation of sequence could be as simple as conserving the 3-dimensional structure of the cleavage site, or another alternative could be as an intracellular part of the cell density signaling process. The role of the larger, N-terminus protein, will require further analysis to resolve why it is so highly conserved.

Our focus has been on molecular characterization of the sensing mechanism for cell density regulation that generates a proliferative signal, but the cell usually requires a more sophisticated response. For instance, in U2OS-pESY1c cells expressing SNZR 1P1L one might postulate that as the cells overgrow the monolayer they begin to produce a secreted inhibitor to prevent the cells in the valleys from producing SNZR 1P1L and dividing: a classic Turing model (Murray, 1988; Turing, 1952). In tendon cells, more is known and with the knowledge of the cofactor one can adjust a previous model to reveal how cells form a growth plate and then fill a growing tube (fibril) with collagen. To do this the cells need a timer, a growth stimulator, a growth inhibitor and a trigger for apoptosis. Mathematical modeling has shown that this can be accomplished with two factors and the cell (Schwarz, 1996). The first factor stimulates growth and the production of the second factor. The second factor, which does not diffuse into the medium; has no activity on its own but by interacting with the first factor changes it to an inhibitor of growth, and a stimulator of procollagen synthesis. Cells left at high cell density cause a drop in the production of the first factor triggering apoptosis. The first factor can now be seen as SNZR 1P2L that we have described. The second factor has until now been hypothetical but a good candidate would be the free cofactor (SNZR 2L). Their interactions occur in the plasma membrane and their consequences are predicted to further organize the membrane structure that in turn can alter the internal state of the cell.

In a permissive cell culture environment for tendon cells – very low serum (0.2%) and addition of ascorbate – these factors come to the forefront and results in the “peculiar” behavior of primary cells in culture (Schwarz, 1991; Schwarz, 1996; Schwarz & Bissell, 1977; Schwarz, Colarusso & Doty, 1976; Schwarz, Farson & Bissell, 1979; Valmassoi & Schwarz, 1988; Zayas & Schwarz, 1992). The cells isolated from 16 day chick embryos were programmed to rapidly make a tendon by way of a growth plate in order that the hatched chick is able to walk to find food and water. In the research laboratory, however, we wanted tendon cells to conform to normal cell culture protocol: grow on a 2-dimensional plastic surface from low density to high density; then be trypsinized off the plastic and the process repeated. Meanwhile the cells were trying to interpret these changes in terms of its own programming for generating a tendon – a collagen rope – by regulating collagen production and cell proliferation. As a result we found that PAT cells only made high levels of procollagen at high cell density (∼50% of total protein synthesis). Left at high cell density for several days, procollagen production declines. Splitting high density cultures resulted in cells that grew to a lower confluent density and produced a lower level of procollagen. Seeding cells at low cell density resulted in uneven cell growth. Seeding cells at low cell density as a small island in the middle of a dish resulted in the whole island apoptosing after a few hours. Gentle agitation of confluent cells seeded as an island in the middle of dish – high medium to cell ratio – caused cells to grow. Adding back CM to an island of confluent cells caused cells to grow. Taking all this information and developing a 2 factor model to explain it was the first step; the second step was to mathematically model the behavior of the cells in 3-dimensional space. Their programming generated a growth plate that was confirmed in vivo at the tendon/muscle junction (Schwarz, 1996).

At the growth plate one can postulate how tendon cells normally respond: at the leading edge cells at moderate cell density are growing driven by their production of the SNZR 1P2L. As one moves towards the trailing edge the cell density increases as does the level of SNZR 2L. This slows proliferation, increases procollagen production and begins a decline in the level of SNZR 1P. At the trailing edge the cells have been at high cell density producing high levels of procollagen for a longer time and the buildup of SNZR 2L causes a continued decrease in the SNZR 1P. This drop eventually triggers apoptosis although some cells at the periphery of the tube (fibril) survive. This is only an outline with the principal players identified (Fig. 15). Nevertheless, a “peculiar” behavior in one setting becomes “normal” in another.

Figure 15 Schematic representation of tube formation – a critical process in diverse tissues.

Glandular tissues clearly need ducts (tubes) to secrete their products. Tendons, a structural tissue, also form tubes but fill them with collagen to make fibrils. This solves the problem of how to make a collagen thread that can be long but maintain a precise diameter for uniform strength. The key element is a growth plate where cells are proliferating and moving forward on one side (right) while they are mostly dying on the opposite side (left). To accomplish this requires two factors: SNZR PL (blue) and SNZR L (dark yellow). Growth is stimulated at moderate cell density by the production by many cells of SNZR PL – the factor is found in the fluid surrounding the cells as well as bound to the plasma membrane. The binding of SNZR PL not only stimulates growth but the production of a non-diffusible second factor that we postulate is SNZR L. This causes another change in the cell membrane and some cells slow growth and become more differentiated which we show as oval cells becoming rectangular in the transition zone. As cells become denser, they accumulate more SNZR PL and SNZR L and become highly differentiated. The exception is cells at the outer edge of the tube that due to their position never reach high cell density (narrow ovals). The high level of SNZR L causes a drop in the production of SNZR P and this triggers apoptosis. In tendon, the differentiated cells produce half of their total protein synthesis as procollagen and they fill the tube with collagen and become a fibril. The cells at the edge are not necessarily contiguous or ordered. One can extrapolate that, in a gland, the differentiated zone would produce a variety of mature phenotypes that could escape apoptosis and form a tight single or multi-cell layered duct.

Analyzing molecular changes raises the question of why does the cell need an additional method for signal transduction? Why not use a protein to activate a receptor? Or put another way, what is the advantage of using a diffusible lipid? In the case of activating a receptor, the cell responds in a graded manner in a set direction. Adding a growth factor can stimulate a cascade of events that results in more rapid cell proliferation. However, the participation of a cell within a growth plate is a far more complex process requiring the cell to respond to parameters related to both space and time. By detecting the levels of SNZR 1P2L and SNZR 2L, the cell can determine its location within the growth plate, whether it should be growing or slowing down, producing high levels of procollagen, or apoptosing. Having a diffusible lipid be a major signal makes the system sensitive to subtle and rapid changes; and with all of the sensors and modifiers acting in one place creates the biological equivalent of an analog logic board allowing the cell to respond to a complex environment correctly.

Key elements to this “circuit” require unique lipids yet this is not a common type. Because they are present at low concentrations, their detection is difficult. We were only able to analyze the two unique lipids because of their tight binding to a protein to which we had both tag and polyclonal antibodies. On the other hand, the tight binding of a protein made purification by standard lipid extraction methods impossible. But the critical observation is that production of a chick protein made in tendon with high homology to the human equivalent causes a unique human bone lipid to be produced in U2OS cells. This makes the protein look like a cassette that can bind a variety of different lipids. But this raises the question of how many kinds are there? If they are in tendon that uses a growth plate to form tubes filled with collagen, do we expect to see unique lipids in other tissues that form tubes – filled (structural tissues) and unfilled (ductal glands) – using a growth plate? Similarly we have seen one case where two cell types share a common protein. But how many cell types share this protein and are there alternative proteins for other cell types? Taking into account the unknown number of unique lipids, one can ask how many unique cell density signals exist in an organism?

Finally, even though our initial aim in selecting an osteosarcoma cell line (U2OS) was simply to use its cellular machinery to express the chicken cell density signal, these cells were not passive responders to the expression of a chicken cDNA. Instead they used this expression to make more of their own cell density signal, and in turn, structures resembling spongy bone with the production of an early osteoblast specific marker. This complex differentiation program was retained by these cells even after becoming malignantly transformed and spending 49 years in cell culture. This indicates that the right signaling molecules can push the malignant cell in new and potentially stable directions. This is not unique to U2OS cells. Using malignant mammary cells (Weaver et al., 1997), “these studies show that despite a number of prominent mutations, amplifications, and deletions, signaling events which are linked to the maintenance of normal tissue architecture are sufficient to abrogate malignancy and to repress the tumor phenotype”. Again, melanoma cells (Postovit et al., 2006) grown on matrices preconditioned by human embryonic stem cells “induced melanoma spheroid formation, promoted the re-expression of Melan-A, and inhibited melanoma cell invasion”. With cell density, the signaling mechanism is beginning to be defined at the molecular level. This raises the question of whether these molecules can be used in a tissue-specific manner to effectively manipulate the differentiated state in order to stimulate the healing of injured tendons and bones, and as a new therapeutic approach to reversing the uncontrolled proliferation of cancer cells by directing cells to normal morphogenesis. The latter would be particularly ironic if studies focused on the differentiated state of tendon cells – the cell type with perhaps the lowest rate of malignancy – provided new insights into controlling this disease. Abbreviations

PAT primary avian tendon

CM conditioned medium

SNZR 1P1L cell density factor with a bone cofactor

SNZR 1P2L cell density factor with a tendon cofactor

SNZR 1P protein factor

SNZR 1L bone cofactor

SNZR 2L tendon cofactor

Supplemental Information

Supplemental Information 1 Broad spectra of the same files used in detail in Fig. 9.

300–750 m/z spectra for three of the files used in Fig. 9. The arrows in the untreated cofactor panel mark the 543.43 and the 557.44 peaks.

Click here for additional data file.

RS would like to thank three people at Amgen for their knowledge, experience and hard work in purifying factors – Bill Kenney, Harvey Yamane, and Mike Bendor. Bill’s encouragement that “this project is hard because all the easy ones have been done” turned out to be true. Eva Lee for pulling lots of embryonic chick tendons, collecting 400 liters of conditioned medium, and working through many problems. The members of the Bissell group past and present who gave advice and encouragement to an eccentric biologist investigating how cells “see” their neighbors. Recently, Ana Correia, Cyrus Ghajar, Jamie Inman, Hidetoshi Mori, Ramray Bhat, Joel Hyman and Sabine Becker-Weiman knowingly and unknowingly helped get this work to the finish line. Finally, Mina Bissell for her contagious enthusiasm for good biology.

We would also like to thank Peter Rademacher for his help in analyzing the mass spectrometry data and giving us many useful interpretations.

Additional Information and Declarations

Competing Interests

Author Contributions

Animal Ethics

Patent Disclosures

Richard Schwarz holds the patent for “Cell density signal protein suitable for treatment of connective tissue injuries and defects (#6433136)” referenced in the Results section. This patent does not and has not generated income and was critical to the completion of this manuscript.

Christopher J. Petzold performed the experiments, analyzed the data, contributed reagents/materials/analysis tools, wrote the paper.

Richard I. Schwarz conceived and designed the experiments, performed the experiments, analyzed the data, contributed reagents/materials/analysis tools, wrote the paper.

The following information was supplied relating to ethical approvals (i.e., approving body and any reference numbers):

The 16 day chick embryos were exempt from requiring an animal use protocol as determined by LBL Animal Welfare and Research Committee. Similarly, the legs from 5 week mice (BALB/c) came from mice already euthanized for other approved protocols and were also exempt.

The following patent dependencies were disclosed by the authors:

Cell density signal protein suitable for treatment of connective tissue injuries and defects/ #6433136/Aug. 13, 2002.

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
