# Peer review of "Cells determine cell density using a small protein bound to a unique tissue-specific phospholipid"

_PeerJ, doi:10.7717/peerj.192_

## Round 0.1 · original submission · Major Revisions

Please pay particular attention to addressing the very significant concerns of Reviewer 1, especially those pertaining to appropriate controls, activity assays and on how you determined the significance of your results.

Reviewer 1 ·

Basic reporting

No comments

Experimental design

not rigorous - key controls missing, no description of repeating experiments
methods missing sufficient detail

Validity of the findings

no statistics
no clear conclusion, other than the experiment didn't work

Additional comments

This manuscript examines a potentially interesting topic – how cells determine their density. Unfortunately, key experiments have not been done, key controls have not been done, and the manuscript appears to have many logic errors. In addition, the writing is very opaque, and much of the writing doesn’t seem to make sense.

Major points
There is no control for the immunofluorescence, and the antibody was used at 1:40. At this high concentration, any antibody will stain any cell or tissue. You need a preimmune control.

Figure 1 shows that two different anti-myc antibodies cannot detect a myc-tagged version of “the protein” made in E. coli. This suggests that you have a frameshift mutation somewhere in the cDNA, and are not making the correct fusion protein. The polyclonal can barely detect the protein used as an antigen- also indicating that something is wrong here.

A variety of purification / modification steps are shown for some sort of protein. However, the critical thing when purifying/ modifying a factor is to show whether it has activity. There are no activity assays done for the various treatments of the factor. You need to do test the activity of a series of concentrations of the ‘factor’, or treated factor, since many signals show activity in a concentration range that has lower and upper limits.

In the mass spectrometry, the lipase buffer control should look pretty much like the untreated material, whereas it actually looks almost exactly like the lipase treatment. Something is clearly wrong here. Two tiny peaks that are different between lipase and lipase buffer control were chosen for analysis, but there are mmany other peaks that could have been similarly chosen. Also, why is only a small region of the mass spectrum shown?

Most importantly, there is no clear demonstration that you have purified or identified the factor affecting tendon cells.

Minor points
Abstract-
“Cell density is the critical parameter controlling tendon morphogenesis” – I imagine many other parameters are also just as critical. Also, what is the evidence for this opening statement?

“with an affinity for the cell layer” what cell layer? A cell layer in the tendon?

“the band that best correlated with a cell proliferation assay” How can a band (what kind of band?) ‘correlate’ with an assay?

“To function as a SNZR would require that the
full length protein be cleaved to a smaller protein, then secreted” Why is this a requirement?

“the chicken cDNA” The cDNA for what?

“to test whether the recombinant protein would exhibit the expected activity” What activity?

“Outside the cell, a small band was detected” What kind of band? Protein? How was it detected?

“Signal transduction is postulated to occur by an increased ordering
of the plasma membrane” Where does this come from? Aren’t most signals sensed by receptors?

Page, line
2,28 “because induction is slow from a single copy gene.” What is the evidence for this? Lots of single copy genes can show a fast induction.
2, 32 “So manipulating procollagen mRNA levels
is not feasible when the cells are required to make high levels of procollagen” This doesn’t make sense.
3,14 “caused a dramatic change” increase? Decrease?
3,23 “The one described and a second that is not diffusible but interacts with
the first and changes the cellular response(Schwarz 1996).” This sentence doesn’t make any sense.
3,29 “This classic approach is complicated by the
finding that the cell type producing this protein binds a unique tissue-specific lipid cofactor
and this composite molecule imparts a tissue-specific response” This sentence is too vague
4,7 give a ref for U2OS, also, why do we need to know “(from frozen stocks that have been in the laboratory for over a dozen years)”?
4,12 if you put 12 ml into a 225 cm2 flask, there will be so little liquid therer the cells will dry up, yes?
4,26 give the primers used
“This yielded an expressed protein
that was 7 AA smaller in the linker between the protein and the tags” where does this come from?

Reviewer 2 ·

Basic reporting

Some of the introduction is too detailed and not relevant to the current research. For example, the discussion of collagen production and regulation of PAT cells could be consolidated.
The introduction should include a discussion of what is known about the gene from which the cell density factor is derived, such as its known cellular functions, cellular location, and other known cleavage forms if they exist.
The nature of the Schwartz (2002) reference should be specified (U.S. Patent?).

Experimental design

The results section could be improved by simply stating what experiments had been conducted and why, rather than introducing each with an extensive discourse on the rationale.
It is not clear exactly what was expressed in E. coli, i.e. which portion of the gene was tagged and expressed. Therefore it is not convincing evidence that the observed larger-than-expected U2OS-expressed protein is aggregated, rather than a longer form with additional amino acid sequences.
It should be clarified why U2OS cells were used as the heterologous expression system, rather than a more standard eukaryotic expression system. Assuming the authors are correct that different cell types express different cofactors, a less committed/differentiated cell type may have been more appropriate.

Validity of the findings

The reader deduces that the straightforward experiment of taking the U2OS conditioned medium, either straight or partially purified, to PAT cells failed to affect their growth. These experiments and results should be more completely described.

Additional comments

This manuscript describes a complex system which is involved in the cellular sensing of cell density. The factor involves a relatively small protein associated with a lipid-based cofactor, a unique situation which has greatly complicated the characterization of this activity.

---

## Round 0.2 · Minor Revisions

The authors are to be commended on a much improved manuscript, with the majority of reviewers’ comments addressed. I believe final polishing of this manuscript could be undertaken by attending to a small number of minor issues before formal acceptance of the paper.

While the biochemical nature of the protein factor, and of the tendon and bone cell cofactors, has been detailed in this manuscript, there appears to be no formal proof per se that either the protein/cofactor or cofactor alone control cell density. Thus, it would be prudent to alter the statement “further proof of its role” in the Abstract to something like “in support of this proposed role…”.

Similarly, could the authors look again at those parts of the manuscript describing the biological role of the protein and cofactors, as the writing here needs to be consistent with the proposed model of controlling cell density. As current written this appears to state explicitly that cell density signaling is controlled by a protein plus cell-specific cofactor.

As the reviewer writes directly, greater prudence is required when ascribing alkaline phosphatase expression to bone production per se, rather than as a phenotypic marker of bone cells.

Please provide a clear reference from the literature in the your manuscript text of the origin, characterization and use of the U2OS cells. The current supporting statement is rather weak i.e.,” from frozen stocks that have been in the laboratory for over a dozen years".

Reviewer 1 ·

Basic reporting

The paper has been improved. The previous version had mislabeled figures, which very much complicated the previous review. The writing is still in places vague, but it has improved from the previous version, which was quite poor. Even the rebuttal letter is odd - here is a section of the response to Reviewer 2:
"One must remember that PAT cells were less than a week earlier in an embryo. They will not be polite like U2OS cells and wait for things to develop."
Polite? How can a cell be polite? To be honest, in many many years of reviewing papers, I have never read anything quite as odd.
Many of the points previously raised have not been answered. Here is just one example: I previously asked the simple question "give a ref for U2OS". This means to give a paper reference for these cells, such as "Smith et al, 1967", and then in the References give the authors, journal name, year, volume, and page numbers. The reference still in the paper is
"(from frozen stocks that have been in the laboratory for over a dozen years)".

Experimental design

no comments

Validity of the findings

no comments

Reviewer 2 ·

Basic reporting

No comments

Experimental design

No comments

Validity of the findings

The majority of this reviewer’s minor comments have been addressed in this revised manuscript. However, some general considerations remain. The biochemical natures of the protein factor, and of the tendon and bone cell cofactors, have been detailed in this manuscript; these findings are valid. Yet, there is no proof that either protein/cofactor or cofactor alone control cell density. Thus, statements such “further proof of its role” in the abstract should be modified to something like “in support of this proposed role…”. Overall the parts of the manuscript describing the biological role of the protein and cofactors need to be rewritten to indicate the results are consistent with the proposed model of controlling cell density, rather than stating explicitly that cell density signaling is controlled by a protein plus cell-specific cofactor.

Additional comments

Lines 287-288: alkaline phosphatase is a marker of bone cells, but does not mean that bone is being produced. The sentence could read “the ridges produced alkaline phosphatase on the cell surface, indicating increased expression of the osteoblast phenotype.”

---

## Round 0.3 · accepted · Accept

Some of the last remaining points have been addressed satisfactorially, and the paper overall now makes a nice contribution to the literature in this area.